# Synergy between the Waste of Natural Resources and Food Waste Related to Meat Consumption in Romania

Teodor Ioan Trasca [1,2,†] , Monica Ocnean [3,†] , Remus Gherman [3,*] , Raul Adrian Lile [4,5] , Ioana Mihaela Balan [3,6,*] , Ioan Brad [3] , Camelia Tulcan [6,7] and Gheorghe Adrian Firu Negoescu [3]

1 Faculty of Animal Productions Engineering and Management, University of Agronomic Sciences and Veterinary Medicine of Bucharest, 011464 Bucharest, Romania; teodor.trasca@usamv.ro
2 Faculty of Food Engineering, University of Life Sciences "King Mihai I" from Timisoara, 300645 Timisoara, Romania
3 Faculty of Management and Rural Tourism, University of Life Sciences "King Mihai I" from Timisoara, 300645 Timisoara, Romania; monicaocnean@usvt.ro (M.O.); ioanbrad@usvt.ro (I.B.); cameliatulcan@usvt.ro (C.T.); gheorghefiru@usvt.ro (G.A.F.N.)
4 Institution Organizing Doctoral Studies, Doctoral School Biotechnical Systems Engineering (ISB), National Polytechnic University of Science and Technology, 060042 Bucharest, Romania; raulile.csm2009@yahoo.de
5 "Aurel Vlaicu" University, 310032 Arad, Romania
6 Research Institute for Biosecurity and Bioengineering, University of Life Sciences "King Mihai I" from Timisoara, 300645 Timisoara, Romania
7 Faculty of Engineering and Applied Technologies, University of Life Sciences "King Mihai I" from Timisoara, 300645 Timisoara, Romania
* Correspondence: remusgherman@usvt.ro (R.G.); ioanabalan@usvt.ro (I.M.B.)
† These authors contributed equally to this work.

**Abstract:** The study examines the dichotomy between individual dietary autonomy and the broader implications of food overconsumption and waste, particularly focusing on meat consumption's environmental, health, and social equity aspects. In the context of increasing awareness about the negative impacts of excessive meat consumption, this research explores the potential benefits of modest dietary shifts, specifically a reduction in animal product intake, on natural resources and the environment. Utilizing data from international and Romanian sources, including data about meat environmental impacts, in original research, the article analyzes the water, carbon, and land use footprints associated with different types of meat, emphasizing the significant differences between beef, pork, chicken, and sheep meat. The findings highlight that even a small reduction in meat consumption, such as 100 g per week per capita, can lead to substantial decreases in water use, carbon emissions, and land use, underscoring the importance of sustainable eating habits. Moreover, the study explores the potential of plant-based proteins as viable nutritional alternatives that can mitigate environmental footprints and foster global food security. Conclusively, this work advocates for a balanced approach that respects individual choices while promoting collective responsibility towards sustainable consumption patterns, emphasizing the role of scientific research and public awareness in driving positive change in dietary habits for environmental conservation and health benefits.

**Keywords:** waste of natural resources; food waste through overconsumption; meat consumption; water footprint; carbon footprint; land use footprint





## 1. Introduction

There are different opinions about people's right to decide what, how much, and how to eat without outside interference. This perspective is based on the concept of individual autonomy and the freedom to make personal decisions about food [1,2]. This is an important theme in discussions of food and consumption and can be seen as part of individual rights as a food security issue [3]. In our society, where the diversity of opinions

and values is a significant feature, these perspectives are expected to exist. Each individual has the right to their own food preferences and choices, and this must be respected [4,5].

However, it is important to recognize that the problem of food overconsumption and food waste has far wider implications than the individual choices of each consumer [6,7]. These include implications for the environment, public health, equitable access to food, and social equity [8,9]. Addressing this issue may involve efforts to raise awareness and education and develop policies that promote responsible eating behaviors without unduly restricting individual freedom. This can be a complex challenge and requires a balance between respect for individual autonomy and collective responsibility towards society and the environment. Open discussions and constructive dialogues can help find solutions that consider both perspectives [6]. It is a sensitive and complex subject, but it is important to continue to explore it and encourage reflection and debate on these matters.

Many of us are aware of our eating habits and admit that we can sometimes fall prey to overeating. Often, this overconsumption is related to foods that bring us comfort or momentary satisfaction. However, awareness of this reality can be accompanied by a feeling of frustration and an inability to stop this behavior, even if we would like to do so. This paradox of awareness and inaction may be familiar to many, but it is often an uphill battle [10].

Many people, especially from high-income countries, are overweight or obese, and their number is increasing. It is a worrying situation, to which the authorities draw attention [11]. This phenomenon has been increasingly acute in recent years, especially in Romania, which shows significant increases in this context. According to the World Health Organization (WHO), approximately 58% of Romanian adults suffer from overweight [12].

For four decades, meat consumption per capita in Romania has been consistently above both the minimum and maximum levels recommended by the EAT-LANCET Commission, with one notable exception. Only in the first year analyzed, 1961, did Romania record meat consumption that was below the maximum limit established by the commission (−9 g/day) but also above the recommended minimum threshold (+34 g/day). These values have increased substantially in the last four decades, reaching from a total meat consumption of 77 g/day in 1961 to 182 g/day in 2021 [13–15] (Figure 1).

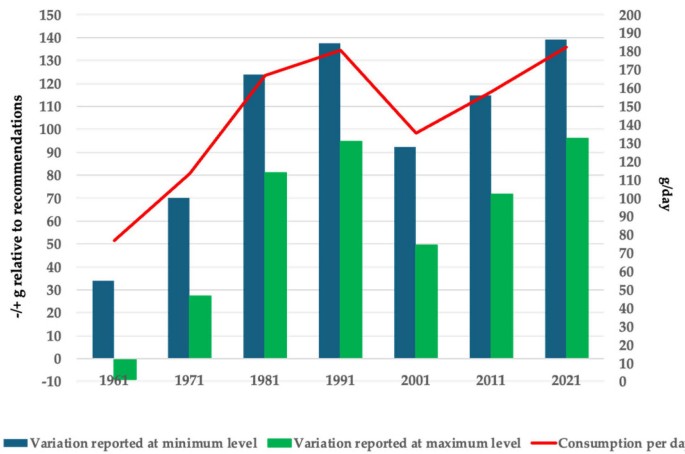

**Figure 1.** Analysis of meat consumption trends in Romania (g/day/person).

This situation reflects a general tendency of the Romanian population to consume meat in larger quantities than those considered optimal, both in the context of human health and in the context of global challenges regarding environmental footprints.

There is, however, a way to address this issue and provide an incentive to reflect and change our eating behavior. Scientific research into nutrition and the impact of food on our health and the environment can play a vital role in this. Scientific studies have brought to light eloquent facts about how excessive consumption of food, especially of animal origin, can contribute to climate change and environmental degradation. Climate

change poses a global challenge that requires collaboration among nations and international organizations [16].

It is crucial to promote international cooperation in developing and implementing solutions to mitigate the impact of food on the environment. Thus, agriculture and food consumption will play a crucial role in meeting this challenge in the context of food security. Climate change is undeniably one of the most significant problems of the 21st century [17–19].

By bringing these scientific data to the fore, we can give consumers a strong reason to think about their food choices. Knowing the real impact and consequences can cause a profound change in our perception of food. Perhaps, when faced with compelling scientific evidence, people will find the motivation to reduce overeating and make more responsible choices [20].

It is important to understand that behavior changes do not always happen overnight and that ongoing support and education are needed. Scientific research can provide not only deeper awareness but also practical solutions and strategies to help people improve their eating habits, which is as a food security issue.

Awareness and scientific research can work together to inspire positive changes in our eating behavior. People have the power to reflect on their habits and make informed decisions for their health and the environment. Knowledge is the key to change, and scientific research gives us this knowledge [21].

One of the most pressing problems in contemporary society is related to the excessive consumption of animal products and its implications for natural resources and the environment. It is a recognized fact that animal products such as meat, dairy, and eggs are the largest consumers of resources compared to plant food sources. At the same time, they are associated with significant risks to human health, especially when consumed in excess [22,23].

In this context, even a modest reduction in consumption of animal products, i.e., 100 g of meat/week, can have a considerable impact on natural resources and the environment because beef production requires huge amounts of water and land and generates high quantities of $CO_2$. Animals raised for meat generate significant greenhouse gas emissions, contributing to climate change [24]. Exploitation of agricultural land for cattle feeding can lead to deforestation and the loss of biodiversity [25].

By reducing the consumption of animal products in general, even through a gradual and moderate change in eating habits, we can help solve these problems. Just one person choosing to eat less beef and more plant-based foods each week can save thousands of liters of water, reduce $CO_2$ emissions, and help reduce pressure on farmland [26].

It is important to highlight that it is not necessary to switch to a vegetarian or vegan diet to have a positive impact. Even small changes, such as adopting a meat-free day or reducing the frequency of meat consumption, such as reducing beef consumption by 100 g per week, can make a big difference in the long term.

Given that animal products are the largest consumers of resources and, at the same time, are associated with risks to human health, any effort to reduce their consumption can help conserve water resources, reduce $CO_2$, and protect agricultural land [20,27]. It is a step in the right direction to find a balance between our nutritional needs and environmental protection.

## 2. Materials and Methods

To carry out the study, we collected, researched, and analyzed second and third-party external data, published by relevant institutions and bodies in the field of statistics, at the international level (Our World in Data—Oxford, UK, FAO—Rome, Italy WHO—Geneva, Switzerland), at the Romanian level (National Institute of Statistics Romania), as well as relevant scientific publications in the field of meat consumption (beef, pork, sheep meat, and chicken meat) and their impact on climate change, which we identified by accessing the most important international scientific databases, i.e., Scopus, Web of Science, PubMed,

ScienceDirect, and Google Scholar. We identified 75 relevant references, of which three are authors' previous publications. We analyzed all these data, and we conducted research regarding the impact of reducing meat consumption in small quantities as part of our research efforts to craft an original article, which serves as the focal point of our study.

This research focuses on identifying the impact of meat consumption on the environment and climate change, starting with the hypothesis that consumer awareness of this phenomenon can play a crucial role in reducing these effects. Through simulations involving even a symbolic reduction in meat consumption—for example, by 100 g per capita per week—the research aims to understand the real impact of this phenomenon on the environment and, implicitly, on climate change. This contribution can serve as a foundation for future scientific research in the field.

It was hoped that, in the future, this research would contribute to consumer awareness by understanding the impact of food overconsumption in general and meat in particular.

Analysis of the water, carbon, and land use footprints for the different types of meat leads to directions for consumption and the development of agriculture in view of climate change and food security [28].

Many Romanian consumers adopt a dietary habit characterized by an excessive consumption of meat, especially beef and pork, which exceeds their nutritional needs. Overconsumption of animal proteins has become a common trend in the Romanian diet [21]. Romanian consumers often tend to associate the quantity of meat with the quality or nutritional value of a meal without considering the negative consequences for health and the environment. This approach can contribute to a range of health problems, such as obesity and chronic disease, and exacerbate pressure on natural resources such as water and agricultural land [29]. Promoting awareness and educating consumers about the importance of balance in their diet and reducing meat consumption can have a significant impact on health and the environment [30].

We considered the following three dimensions of ecological footprint associated with meat consumption (beef, chicken meat, pork, and sheep meat):

- water footprint,
- carbon footprint,

    and

- land use footprint.

To properly assess environmental impact, we analyzed the water footprint, which represents the amount of water consumed in meat production (Water Footprint Network), the carbon footprint [31,32], which measures conventional greenhouse gas emissions generated by this production, and the land use footprint [32,33], which assesses the area of land required for raising animals intended for meat production for 100 g of each type of meat.

The relationship between water, carbon, and land use footprints and climate change is a complex and interconnected issue [34]. These footprints serve as indicators of the environmental impact resulting from human activities, particularly those in the food industry.

Water footprint refers to the volume of water used in food production, an essential resource that, when overused, can lead to the depletion of freshwater reserves and the degradation of aquatic ecosystems. Climate change is impacted by alterations in the hydrological cycle. For instance, deforestation to expand crops can lead to rapid water runoff, triggering both floods and droughts. This imbalance in water dynamics can influence climate patterns locally [35].

The carbon footprint is a measurement of greenhouse gas emissions, such as carbon dioxide ($CO_2$), methane ($CH_4$), and nitrous oxide ($N_2O$), produced by food production. Agriculture, especially animal husbandry for meat production, is a major contributor to greenhouse gas emissions. Cattle farming, for example, emits large amounts of methane, a more potent greenhouse gas than $CO_2$, leading to global warming and climate change [16].

The term land use footprint defines the area of land required for food production. Deforestation and the conversion of natural habitats into farmland have significant effects

on the environment and climate change. Land-use change can contribute to the loss of biodiversity and the release of carbon stored in vegetation and soil, exacerbating the greenhouse effect [36].

Usually, these footprints are significantly related (Figure 2). For example, red meat production, which results in a large carbon footprint due to methane emissions from the animal's digestive system, is also linked to substantial water consumption and extensive land use for animal feeding. Reducing meat consumption and transitioning to a plant-based diet can effectively reduce water, carbon, and land footprints.

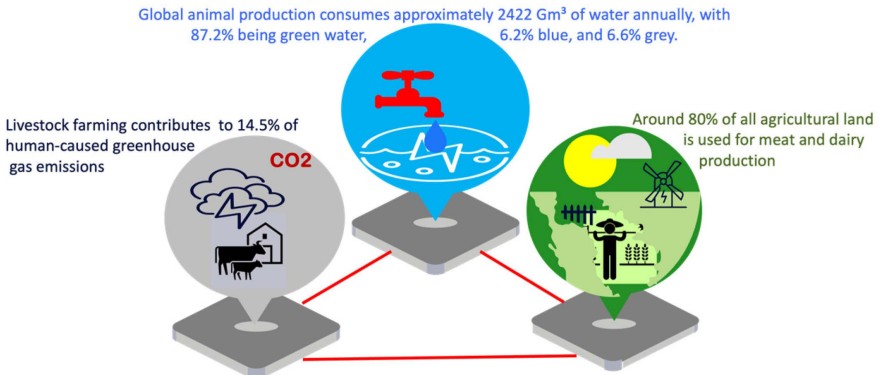

**Figure 2.** Environmental impact of meat production. Source: Authors' interpretation of [37,38].

The most important footprints of meat (beef, pork, sheep meat, and chicken meat) were analyzed, looking at how climate change influences even a minimal reduction in food consumption. In this context, we calculated how it acts on determinants of climate change, i.e., a reduction of 100 g/week of meat in the meat consumption of the adult population in Romania, because overconsumption of food has determined that the adult population in Romania includes meat in particular. Corroborating these aspects, we analyzed how this reduction may affect natural resources.

Climate change is deeply influenced by food production. Water, carbon, and land use food footprints have a significant impact on climate change through natural resource use and greenhouse gas emissions [39].

*Water footprint.* Linked to changes in the hydrological cycle, the water footprint of food products is related to water consumption in agriculture and processing. This use can change the availability of water in regions, affecting the hydrological cycle and causing water shortages or floods. These changes can, in turn, influence rainfall patterns and groundwater levels, with significant implications for local and regional climate [35].

Related to the cooling effect, the transpiration process of plants, vegetation, and crops contributes to cooling air in agricultural areas. Changes in areas used for agriculture and types of crops can affect this effect, influencing local and regional temperatures.

The water footprint of a product consists of three distinct categories: blue water footprint (represents the amount of surface and underground water used in production of a good or service, which does not return to original source), green water footprint (refers to water from precipitation that does not drain or recharge groundwater, but is stored in soil or remains temporarily on soil or vegetation, later evaporating or transpiration through plants) and gray water footprint (refers to freshwater pollution associated with production of a product throughout the supply chain, representing the volume of water required to assimilate pollutants and maintain water quality above agreed quality standards) [35].

The formula that analyzes the Water Footprint Reduction Factor Based on Meat Type and Time ($WFR_t$) is the method developed and utilized in this research for estimating the reduction of water footprint associated with meat consumption, considering both the type of meat consumed and the time period. This formula is important in researching the impact of meat consumption on water resources and climate change, providing a practical

way to assess potential reductions in meat consumption and, implicitly, the impact on the environment.

$$WFR_t = WF_t \cdot mq_t \cdot w_n \qquad (1)$$

$WFR_t$ = water footprint personal reduction depending on the type of meat (liters);
$WF_t$ = water footprint of the type of meat (liters/kg);
$mq_t$ = meat quantity from the type analyzed (kg);
$w_n$ = number of weeks.

The presented formula reflects the calculation of personal water footprint reduction ($WFR_t$) depending on the type of meat consumed. It takes into account three main elements: the water footprint specific to the type of meat ($WF_t$), the amount of meat consumed of that type analyzed ($mq_t$), and the number of weeks ($w_n$) over which the analysis spans. Essentially, the formula multiplies the water footprint of the type of meat (liters/kg) by the amount of meat consumed (in kilograms) and the number of weeks, thus providing a measure of personal water reduction through meat-related food choices.

By applying this formula, we estimated the reduction in water footprint that can be obtained based on the type and quantity of meat consumed over a certain period of time. This approach allowed us to evaluate the potential impact of reductions in meat consumption on water resources.

*Carbon footprint.* Food production, processing, and transport generate significant emissions of $CO_2$ and other greenhouse gases. These emissions directly contribute to global warming and climate change. Agriculture can significantly contribute to reducing greenhouse gas emissions and conserving natural resources by diversifying and adapting production methods that exploit species [40].

Reducing the $CO_2$ footprint of food products through more sustainable agricultural practices, efficient transport, and waste management can help reduce the impact on the climate [30,40].

The formula for analyzing the Carbon Footprint Reduction Factor Based on Meat Type and Time ($CO_2eFR_t$) is a method that was elaborated and employed in this research to estimate the carbon footprint associated with meat consumption. It considers both the type of meat consumed and the duration of consumption.

$$CO_2eFR_t = CO_2eF_t \cdot mq_t \cdot w_n \qquad (2)$$

$CO_2eFR_t$ = carbon footprint personal reduction depending on the type of meat (kg);
$CO_2eF_t$ = carbon footprint of the type of meat (kg);
$mq_t$ = meat quantity from the type analyzed (kg);
$w_n$ = number of weeks.

The presented formula calculates the personal carbon footprint reduction ($CO_2eFR_t$) depending on the type of meat consumed. It considers three key variables: the carbon footprint associated with the specific type of meat ($CO_2eF_t$), the amount of meat of that type that is consumed ($mq_t$), and the number of weeks ($w_n$) over which the assessment is made. Simply put, the formula multiplies the carbon footprint per kilogram of meat (in kg $CO_2e$) by the amount consumed of that type of meat (in kilograms) and the number of weeks, resulting in a number that represents your personal carbon footprint reduction through food choices specific to meat consumption.

By utilizing this formula, we estimated the reduction in carbon footprint based on the type and quantity of meat consumed within a specified time frame. This methodology enables an evaluation of potential reductions in carbon emissions associated with changes in meat consumption patterns.

*Land use footprint.* The expansion of agricultural land and pastures can lead to the clearing of forests and other natural ecosystems, changing land use. This can release carbon stored in soil and vegetation, contributing to an increase in $CO_2$ concentration in the atmosphere. Changes in land use can also alter the albedo (reflectance) of land, which influences the absorption of solar radiation and thus regional temperatures [26].

For this research, we elaborated and utilized another formula for determining the Land Use Footprint Reduction Factor Based on Meat Type and Time ($L_uFR_t$) to estimate the reduction in land use footprint associated with meat consumption. It takes into account both the type of meat consumed and the duration of consumption.

$$L_uFR_t = L_uF_t \cdot mq_t \cdot w_n \tag{3}$$

$L_uFR_t$ = land use footprint personal reduction depending on the type of meat ($m^2$);
$L_uF_t$ = land use footprint of the type of meat ($m^2$/kg);
$mq_t$ = meat quantity from the type analyzed (kg);
$w_n$ = number of weeks.

The formula shown calculates your personal land use footprint reduction ($L_uFR_t$) based on the type of meat consumed. It includes three essential components: the land use footprint specific to each type of meat ($L_uF_t$), the amount of meat of that type consumed ($mq_t$), and the number of weeks ($w_n$) over which the calculation is made. Basically, the formula multiplies the land use footprint per kilogram of meat (in square meters) by the amount of meat of that type consumed (in kilograms) and the number of weeks, providing a measure of your personal land use footprint reduction through your diet related to meat consumption.

By applying this formula, we estimated the reduction in land use footprint based on the type and quantity of meat consumed over a specified period. This approach facilitates an assessment of potential reductions in land use associated with changes in meat consumption patterns.

## 3. Results

We researched the impact of reducing meat consumption by 100 g per week for one year on climate change, analyzing exclusively the adult population in Romania, aged over 18 years (Table 1).

**Table 1.** Resident population in Romania (underage and adults) [27].

| Category | Total Resident Population | Resident Population % |
|---|---|---|
| 0–18 y.o. | 4,112,985 | 21.59 |
| 18–over 85 y.o. | 14,940,830 | 78.41 |
| Total | 19,053,815 | 100 |

We chose to conduct this study only on the adult population over 18 years of age in Romania, taking into account the fact that it is a segment of the population that no longer requires the development of the body based on high animal protein support. This was the reason why the population of children under 18 was not considered. Of course, just as there are adults who do not overconsume meat in general, in the same sense there are also children who overconsume meat, which is why they could have been included in this research. However, the purpose of this research was not to quantify the overconsumption of meat but to present the idea of how the reduction of 100 g of meat per week in human consumption impacts the environmental footprints of the different types of meat consumed by the Romanian population.

### 3.1. Water Footprint of Meat

We analyzed the total water footprint and presented its three components (blue water, green water, and gray water). However, we consider that the total water footprint is one that is the most representative in the context of research on the impact of the four types of meat (beef, chicken meat, pork, and sheep meat) on the environment because it provides overall information (Table 2).

**Table 2.** Water footprint of different types of meat [36].

| Water Footprint | Beef | Chicken Meat | Pork | Sheep Meat |
|---|---|---|---|---|
| Total water footprint liters/kg meat of which: | 15,415 | 4325 | 5988 | 10,412 |
| Blue water % | 94 | 82 | 82 | 94 |
| Green water % | 3 | 7 | 8 | 5 |
| Grey water % | 3 | 11 | 10 | 1 |

By analyzing the data in the table, important aspects can be revealed about the differences in the water footprint levels of the four types of meat. Related to a normal portion of 100 g of meat, the water footprint will show significant variations from one species to another (Table 3).

**Table 3.** Water footprint related to 100 g of different types of meat (liters) *Sources:* Authors' interpretation of [36].

| Water Footprint L/100 g Meat/Week/Capita | Beef | Chicken Meat | Pork | Sheep Meat |
|---|---|---|---|---|
| Blue water | 1449.01 | 354.65 | 491.02 | 978.73 |
| Green water | 46.245 | 30.275 | 47.904 | 52.06 |
| Grey water | 46.245 | 47.575 | 59.88 | 10.41 |
| Total Water | 1541.5 | 432.5 | 598.8 | 1041.2 |

Chicken meat requires about 75.54% less blue water than beef. At the same time, chicken meat uses about 34.37% less green water than beef and has about 2.14% less gray footprint than beef. Overall, the water footprint of chicken is approximately 71.94% lower than that of beef.

Pork has an approximately 64.16% lower blue water footprint than beef. Pork also uses about 26.35% less green water than beef and has about 17.85% less gray water footprint than beef. In total, the water footprint of pork is approximately 61.18% lower than that of beef.

Sheep meat has a 49.39% lower blue water footprint than beef. It requires about 8.93% less green water than beef and has about 82.28% less gray water footprint than beef. In total, the water footprint of sheep meat is approximately 32.42% lower than that of beef.

Applying the $WFR_t$ formula (Formula (1)) to all types of meat analyzed and to the entire adult population in Romania, it is revealed that huge amounts of water can be reduced by simply reducing meat consumption by 100 g/capita/week (Table 4).

**Table 4.** Water footprint related to 100 g of meat weekly/capita/year (million cubic meters (MCM)) Sources: Authors' interpretation of [36].

| Water Footprint of 100 g Meat Weekly/Total Romanian Adult Population/Year | Beef | Chicken Meat | Pork | Sheep Meat |
|---|---|---|---|---|
| Blue water footprint of 100 g of meat weekly/total Romanian adult population/year | 1125 | 276 | 381 | 760 |
| Green water footprint of 100 g of meat weekly/total Romanian adult population/year | 36 | 24 | 37 | 40 |
| Grey water footprint of 100 g of meat weekly/total Romanian adult population/year | 36 | 37 | 47 | 8 |
| Total water footprint of 100 g of meat weekly/total Romanian adult population/year | 1197 | 336 | 465 | 808 |

These results highlight the significant water footprint differences between the four types of meat analyzed and beef, highlighting the reduced impact of chicken, pork, and sheep meat consumption on water resources compared to beef (Figure 3). Reducing beef

consumption and switching to less water-intensive alternatives can significantly contribute to the conservation of this important resource.

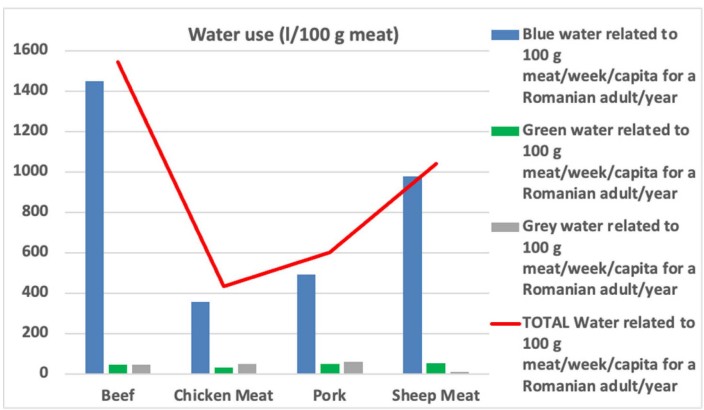

**Figure 3.** The impact on the water footprint of reducing weekly consumption by 100 g of meat per capita/year.

### 3.2. Carbon Footprint of Meat

Regarding the carbon footprint analyzed by carbon dioxide emission ($CO_2$e), significant differences in $CO_2$e can be observed. Analyzing the data in Table 5, the percentage differences between the four types of meat in terms of carbon footprint are revealed.

**Table 5.** Carbon footprint related to 100 g of different types of meat (kg). Sources: Authors' interpretation of [15].

| Carbon Footprint $CO_2$e | Beef | Chicken Meat | Pork | Sheep Meat |
|---|---|---|---|---|
| $CO_2$e kg/kg meat | 144.0 | 10.7 | 14.3 | 186.0 |
| Total Kg $CO_2$e related to 100 g meat/week/capita | 748.80 | 55.64 | 74.36 | 967.20 |

The analysis of the carbon footprint associated with meat consumption reveals significant differences between the types of meat analyzed. In particular, sheep meat stands out for having the largest carbon footprint, both in terms of $CO_2$ equivalent emissions per 100 g of meat and in terms of the annual per capita impact associated with a weekly consumption of 100 g of meat for an adult from Romania.

Specifically, sheep meat has a carbon footprint of 18.60 kg $CO_2$e per 100 g of meat, which is significantly higher compared to the other types of meat analyzed. Beef, with a footprint of 14.40 kg $CO_2$e per 100 g, is approximately 22.58% lower than that of sheep. However, the impact is much lower for chicken and pork, with footprints of 1.07 kg $CO_2$e and 1.43 kg $CO_2$e per 100 g of meat, respectively. Compared to sheep meat, the carbon footprint of chicken is 94.25% lower, while that of pork is 92.31% lower, making them much more sustainable options from the perspective of greenhouse gas emissions.

When looking at the annual per capita impact, based on a weekly consumption of 100 g of meat, sheep meat remains the type of meat with the highest impact, generating 967.2 kg $CO_2$e. In contrast, beef generates 748.8 kg $CO_2$e, being 22.57% less impactful than sheep meat. Similarly, chicken and pork present much reduced values, with 55.64 kg $CO_2$e and 74.36 kg $CO_2$e, respectively. This further emphasizes that chicken and pork are alternatives with a significantly lower environmental impact compared to sheep meat and beef, in the context of regular consumption in Romania.

Applying the formula $CO_2eFR_t$ (Formula (2)) and reporting the carbon footprint related to the entire adult population in Romania, these values of reducing meat consumption by 100 g/week/per capita over a whole year become even more significant (Table 6).

**Table 6.** Carbon footprint related to 100 g of meat weekly/total Romanian adult population for one year (tonnes) Sources: Authors' interpretation of [15].

| Specification | Beef | Chicken Meat | Pork | Sheep Meat |
|---|---|---|---|---|
| Total $CO_2e$ footprint of 100 g of meat weekly/total Romanian adult population/year | 38.93 | 2.89 | 3.867 | 50.29 |

These data could have important implications for sustainability and food policy debates, prompting a reflection on our consumption choices and how they affect the environment through their carbon footprint. Reducing the consumption of meat with a high carbon footprint could be an effective strategy to mitigate the impact of climate change. Alternatively, opting for meats with lower emissions, such as chicken and pork, can contribute to a more sustainable consumption pattern (Figure 4).

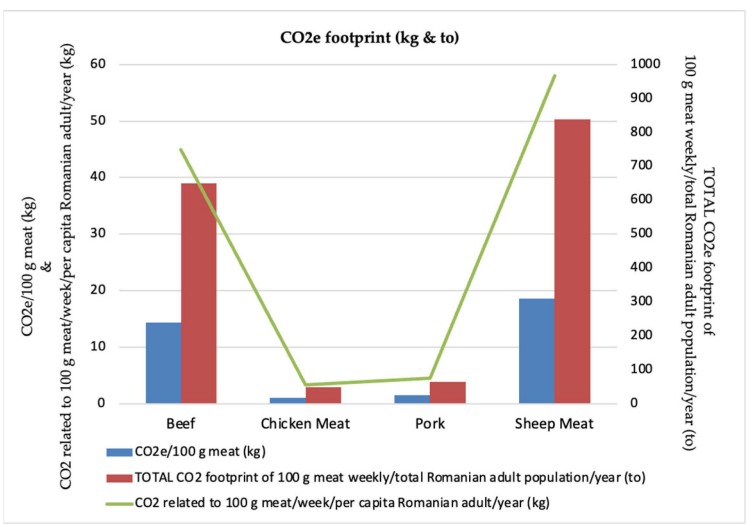

**Figure 4.** The impact on the carbon footprint of reducing weekly consumption by 100 g of meat per capita/year.

These results underscore the importance of choosing foods with a lower carbon footprint to reduce the impact of climate change. Eating chicken meat is a greener option compared to beef or sheep meat from a carbon perspective. Reducing red meat consumption and switching to alternatives that use fewer greenhouse gases can make a significant contribution to reducing the carbon footprint of a diet.

*3.3. Land Footprint of Meat*

Analyzing the data in Table 7, significant aspects emerge regarding the percentage differences between the four types of meat in terms of land use.

**Table 7.** Land use footprint related to 100 g of different types of meat. Sources: Authors' interpretation of [34].

| Land Use Footprint $m^2$ | Beef | Chicken Meat | Pork | Sheep Meat |
|---|---|---|---|---|
| Land $m^2$/kg meat | 326.2 | 12.2 | 17.4 | 369.8 |
| Total land $m^2$ related to 100 g of meat/week/per capita/year | 1696.29 | 63.54 | 90.27 | 1923.01 |

Based on the data provided in Table 7 regarding land use footprint, it is evident that different types of meat production have varying impacts on land resources compared to sheep meat, which serves as the reference point due to having the highest land use footprint.

When examining land use per 100 g of meat, chicken meat and pork both require a relatively small fraction of the land needed for sheep meat, approximately 3.31% and 4.69%, respectively. In contrast, beef has a significantly higher land use footprint, utilizing a substantial portion, around 88.18%, of the land required for sheep meat.

Regarding the total land use related to meat consumption on a weekly basis per capita per year, similar trends are observed.

These comparisons highlight significant disparities in land utilization for meat production, with chicken and pork showing much lower land use footprints compared to sheep meat and beef. Such insights are crucial for understanding the environmental impacts of different meat types and informing sustainable consumption practices.

By employing the formula $L_uFR_t$ (Formula (3)) and evaluating the land use footprint across the entire adult population of Romania, the implications of decreasing meat consumption by 100 g per week per capita annually gain further prominence (Table 8).

**Table 8.** Land use footprint related to 100 g of meat weekly/total Romanian adult population for one year (kha). Sources: Authors' interpretation of [34].

| Specification | Beef | Chicken Meat | Pork | Sheep Meat |
|---|---|---|---|---|
| Total land footprint of 100 g of meat weekly/total Romanian adult population/year (kha) | 2534 | 95 | 135 | 2873 |

These findings highlight the importance of choosing foods with a smaller land footprint to reduce the impact of climate change and the pressure on terrestrial ecosystems. Reducing consumption of sheep meat and beef in favor of chicken meat or other more land-use-efficient protein sources can make a significant contribution to protecting and conserving agricultural and natural land (Figure 5).

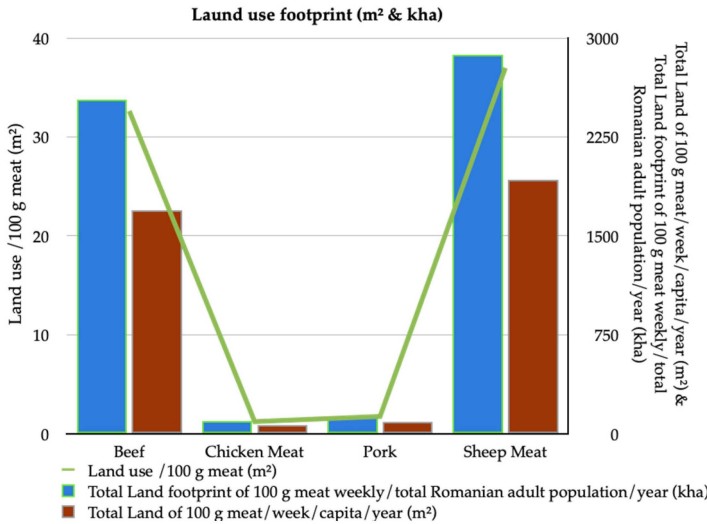

**Figure 5.** The impact on the land use footprint of reducing weekly consumption by 100 g of meat per capita/year.

## 4. Discussion

In light of awareness about environmental issues and the pressing need for sustainable eating habits, we find ourselves facing a critical question: What are the environmental and resource costs of meat consumption in Romania? In a society where consuming meat is often intertwined with cultural and dietary norms, it is crucial to scrutinize the environmental consequences and the resources expended in meat production and consumption [41]. Romania, like many other countries, faces significant challenges regarding

the environmental impact of meat consumption. From deforestation for agricultural land dedicated to livestock farming to greenhouse gas emissions generated by the meat industry, the impact is considerable. This examination of the real costs of meat must include an assessment of all these factors to obtain a complete picture of the consequences for the surrounding environment.

Amid escalating global consciousness about ecological preservation and the push for dietary sustainability, the environmental and resource implications of Romania's meat consumption warrant a deeper exploration. This consideration is pivotal, especially in a societal framework where meat is central to cultural and dietary identities. Analyzing the environmental toll and resource demand linked to meat's lifecycle becomes imperative. Furthermore, integrating insights from various international studies highlights the universal urgency to reassess meat consumption patterns, positioning Romania's situation within a global narrative that calls for a concerted move towards sustainable food practices to address global environmental challenges [42–45].

Another important aspect is the utilization of resources associated with meat production. Meat consumption requires significant amounts of water, agricultural land, and feed for animals raised for human consumption. These resources are finite, and in many cases, their use for meat production may be considered inefficient in terms of sustainability. In a world where natural resources are increasingly limited, it is essential to evaluate how we use these resources and seek more sustainable alternatives [26].

Examining the real cost of meat in Romania also requires careful consideration of the social and economic aspects. While focusing on assessing the real ecological cost of meat's environmental footprint in Romania, it is essential to also consider the effects on local farming communities, where raising animals for meat can be a key source of income. Also, the economic accessibility of plant-based alternatives for Romanian consumers and the potential consequences of changes in agricultural subsidies and environmental policies are important. All these social and economic elements intertwine with environmental issues, and a comprehensive understanding of them is crucial for formulating effective strategies to mitigate the environmental footprint of meat without negatively affecting the well-being of the population or the economic stability of the country. Although meat consumption may be an important part of the economy and culture in some communities, it is important to question whether the economic and social benefits justify the ecological and resource costs associated with meat production and consumption. Research indicates that transitioning to a more plant-based diet could have significant benefits for both the environment and human health, and this aspect should be taken into account in discussions about the future of food in Romania and worldwide [46–49].

Examining the ecological impact and resource usage associated with meat production and consumption in Romania is essential to truly understanding the real cost of this food. Cattle numbers in Romania decreased from year to year. In 1990, Romania had over five million head of cattle, but every year the herds decreased nationally, so now it has less than one million head. According to the Romanian Ministry of Agriculture, Romania imports 70% of beef meat and 60% of pork consumed nationally annually, with domestic production ensuring 30% and 40% of consumption, respectively. At the same time, in Romania, there is a degree of self-sufficiency of 150%, which means that 50% of the chicken meat produced in Romania is exported. Romania is also one of the most important exporters of sheep meat, especially outside the EU [50]. The excessive consumption of meat has a significant impact on the water footprint, representing a major concern for environmental sustainability, including in the context of Romania. The processes involved in meat production require large quantities of water, such as animal husbandry, irrigation of crops for their feed, and meat processing [26]. In addition to the substantial pressure that excessive meat consumption exerts on water resources in Romania, this problem reflects a global challenge. Globally, the agricultural sector, and especially meat production, is responsible for massive freshwater consumption, intensifying competition for limited water resources and contributing to water stress in many regions. By reducing reliance on meat

and adopting more plant-based diets, consumers can help reduce the water footprint of the human diet, addressing one of the most pressing sustainability issues and promoting more efficient use of water resources globally [28,35].

Thus, excessive meat consumption contributes to the depletion of water resources and water pollution through the use of pesticides and fertilizers in feed crops. Reducing meat consumption can have a significant impact on the water footprint and can contribute to the conservation and protection of this vital resource for life on Earth, including within the specific context of Romania. By reducing beef consumption by 100 g of meat per week/total Romanian adult population/year, the volume of water equivalent to the water footprint related to this total amount of beef (1197 MCM) is approximately equal to the water volume of the largest accumulation lake in Romania, the well-known Bicaz Lake (1250 MCM), an international tourist attraction [13] (Figure 6).

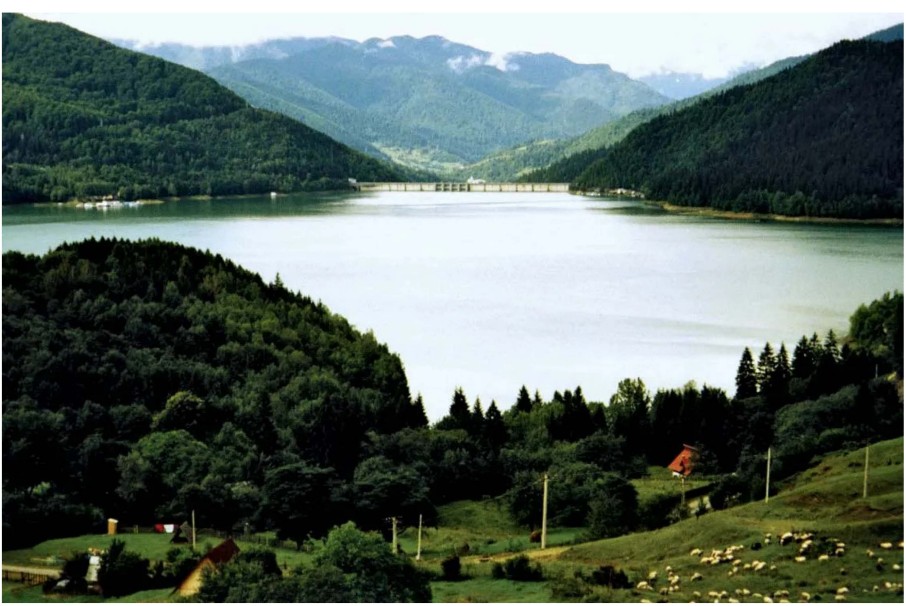

**Figure 6.** Bicaz Lake [51].

Carbon footprints fluctuate based on various factors, such as production location, energy source, transportation distance, and more. Consequently, the carbon footprint of a specific product varies [52].

The values presented offer a broad overview, indicating general trends rather than precise calculations for specific instances ($CO_2$ Everything). For analyzing the same meat type, respectively beef, according to the $CO_2$ Everything website, the carbon footprint of a 100 g portion of meat is similar to that of 78.7 km of driving a car; respectively, each kilometer of driving a car generates 0.2 kg of $CO_2$ (more precisely, 0.196950445 kg $CO_2$/km, on average for a small car). Relating this value to 748.80 kg of $CO_2$ related to a consumption of 100 g of beef/week/capita/year results in a transposition of 3744 km of car driving annually for each adult in Romania. This distance covers the distance from Bucharest to Lisbon, Portugal, one of the most distant capitals in the EU, compared to the capital of Romania [53]. If a family consisting of children and two adults travels this way back and forth by car, the carbon footprint of this road is equivalent to the carbon footprint of the reduction of beef consumption by 100 g/week/year related to the two adults (Figure 7).

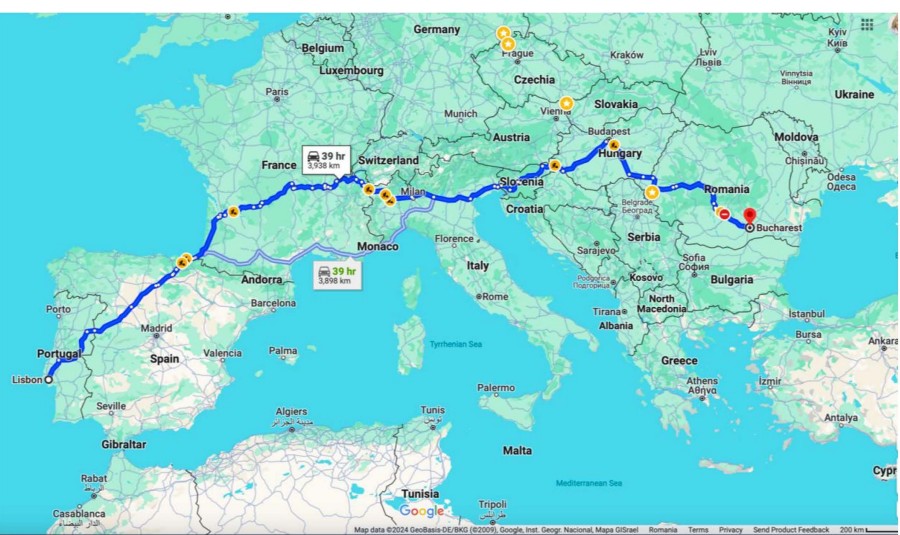

**Figure 7.** The distance between Bucharest, Romania, and Lisbon, Portugal [53].

Land use footprint is an essential aspect in analyzing the impact of meat consumption, and the data presented highlights the importance of reducing beef consumption. By reducing consumption by 100 g per week per capita over a year, we observe a significant reduction in the land use footprint. For example, for beef, this reduction would be equivalent to approximately 0.17 hectares. To contextualize this figure, it is important to consider Romania's agricultural land area, which is approximately 13.5 million hectares, according to the European Commission's report [54]. At the same time, we have an adult population of approximately 14.94 million people in the country [27].

These data provide us with a clear perspective on the long-term impact of reducing beef consumption. For example, in approximately 5 years, the land use footprint associated with this reduction could equal Romania's entire agricultural land area. This underscores the importance of sustainable management of natural resources and how our consumption choices can influence agricultural land use and ecological balance.

The study does not present data on meat consumption by children and adolescents. Although we considered that it is not recommended to propose reducing meat consumption among Romanians under 18, in fact, the situation can be similar, a conclusion we draw considering that a large percentage of them are overweight and obese [12]. The shortcoming of this study lies in the fact that the reduction of meat consumption in this category was not considered. However, it is quite risky to recommend reducing meat consumption to people under 18, because most of them have increased needs for growth and development, being in the period of life when the body requires a specific diet, especially one rich in protein. In the same context, it is well known, both by parents and by dieticians, that children and adolescents are often difficult to convince to eat certain vegetables, which is why animal products, especially meat, are an important source of protein in their food [55,56].

In examining the environmental impact of meat consumption in Romania, it is important to distinguish between estimates based on average unit values and the actual impact, given the significant differences determined by local production specificities and the complex nuances of the supply chain. These differences highlight the need for a detailed and nuanced assessment to understand the true ecological burden associated with different types of footprints.

The water footprint of meat consumption can vary considerably, depending on local water management, use efficiency, and specific farm conditions. The average values do not take into account these local differences, which may lead to an underestimation or overestimation of the actual water resources needed in certain areas of Romania. Local production conditions, such as water and grassland management, can contribute to a smaller water footprint compared to places where these practices are less efficient [23,26,29,36,42,44,57].

The carbon footprint shows significant variations that can be masked by average unit values. The significant import of beef, especially from Spain and Italy, underlines the fact that a large part of the beef consumed in Romania is imported, which can significantly increase the carbon footprint. The distance that this imported meat has to travel to reach Romanian consumers' tables adds additional $CO_2e$ throughout the product's life cycle. Long-distance transport or energy-intensive methods, whether by road, rail, air, or sea, contribute significantly to greenhouse gas emissions. In addition, production methods and environmental regulations vary significantly between exporting countries and Romania, directly influencing the carbon footprint of the meat consumed [23,26,29,42,58] (Table 9).

**Table 9.** $CO_2e$ comparison for different freight modes over 1000 km and their driving equivalents [58].

| Specification | CO$_2$e for 1000 km/kg Meat | Equivalent km of Driving |
|---|---|---|
| Freight—Air | 2.210 | 11.20 |
| Freight—Road/Truck | 0.105 | 0.55 |
| Freight—Rail | 0.025 | 0.15 |
| Freight—Shipping | 0.015 | 0.10 |

The beef consumed in Romania is imported mostly from Spain (road distance from Madrid to Bucharest—3181 km) and Italy (road distance from Rome to Bucharest—1876 km), or it is produced locally, but it is usually freighted by refrigerated trucks. Therefore, the calculations regarding the $CO_2e$ footprint are made on the total beef carcass, freighted by truck. This practice reflects a global trend, adapting the meat-cutting method to the needs of the local market (Table 10).

**Table 10.** Comparative $CO_2e$ footprint of beef freight on international road routes from Spain, Italy, and domestic road routes in Romania [15,58].

| Specification | CO$_2$e for Beef Imported from Spain (g/kg) | CO$_2$e for Beef Imported from Italy (g/kg) | CO$_2$e for Beef from Romania (300 km Distance from Farm to Fork) (g/kg) |
|---|---|---|---|
| CO$_2$e—freight to Romania by road (truck) | 0.33 | 0.20 | 0.03 |
| CO$_2$e kg/kg beef—production | 144 | 144 | 144 |
| Total CO$_2$e g/kg (production + freight) | 144.33 | 144.20 | 144.03 |
| Total Kg CO$_2$e related to 100 g of meat/week/capita | 750.54 | 749.82 | 748.96 |

After import, the beef carcasses are cut in Romania, where they are processed into various products (salami, sausages, etc.) or sold as cutting pieces, ready for household consumption or use in public food. In this context, the $CO_2e$ footprint may increase due to related freight.

Likewise, the land footprint must consider not only the space used for grazing or forage production but also impacts on land degradation, deforestation, and biodiversity loss, which may not be fully represented in average unit values. Land use in Romania for meat production may be comparatively more or less intensive compared to other countries, causing discrepancies between actual and estimated figures [23,26,29,42].

Therefore, while average unit values provide a basis for understanding the ecological costs of meat consumption, detailed and localized data are essential to assessing the true size of the environmental impact in Romania. A deeper and more localized understanding can inform sustainability policies and practices in a more targeted and effective way, highlighting the need for measures adapted to national and regional specificities.

For consumers who do not overconsume meat, an important solution is the replacement of proteins from meat, such as beef, with alternative protein sources. At the same time, by reducing meat consumption or substituting it with other types of animal or plant-based protein sources, consumer budgets will be positively affected because meat in general,

but especially beef, is one of the most expensive foods in the Romanian market. Table 11 provides comparisons between different animal-based and plant-based protein sources, highlighting the specific amounts of plant foods needed to equal the protein content of certain animal foods.

**Table 11.** Protein content: animal-based vs. plant-based foods [59].

| Type of Food | Protein Equivalent | Protein |
|---|---|---|
| One beefburger | 175 g cooked lentils | 18 g |
| One egg | 78 g oats | 13 g |
| Chicken breast (80 g) | 210 g cooked chickpeas | 21 g |

The data highlights the versatility of plant-based diets in providing protein and reveals plant-based alternatives to traditional animal protein sources, both for health benefits and sustainability considerations.

However, price considerations must also be factored in. With the importation of a part of the necessary beef, beef prices in the Romanian market have seen a considerable increase in recent years [60]. Consequently, given the cross-price elasticities in the case of substituting beef with another source of protein, there may be a surge in demand for alternative sources of animal-based proteins, particularly chicken meat and pork, leading to their price escalation [8,61]. The most accessible alternatives, both economically and in terms of consumer health, include non-meat, animal-based protein sources such as eggs and dairy products. Plant-based protein sources, known for their significantly lower prices, offer the greatest benefits to environmental sustainability, consumer health, and household budgets.

## 5. Conclusions

The debate about individual food freedom versus collective responsibility focuses on balancing personal choice with the larger impact of overconsumption and food waste on societal health. Awareness is growing of the health, environmental, and animal welfare disadvantages of high meat consumption, leading to greater acceptance of plant-based protein as a sustainable alternative [26,62–64].

Data indicate that ruminant meat has a substantial impact on the environment, consuming vast resources and contributing to greenhouse gas emissions, deforestation, and pollution. Scientific findings on the environmental costs of meat may prompt a shift towards greener food choices [17,65].

The presented model can be adapted and applied in any country, regardless of its level of development or cultural specificities. This could be one of the potential solutions to global issues concerning environmental footprints, particularly addressing the situation of overconsumption of meat, a situation that Romania finds itself in, by reducing the weekly consumption by just 100 g, preferably ruminant meat. Regarding protein sources, there has been a significant evolution in how we view human nutrition and how we relate to animal versus plant proteins [17,65].

The analysis shows significant differences in the environmental impact of meat, with chicken and pork being less harmful than beef and sheep meat. Additionally, methane from ruminants greatly accelerates global warming, highlighting the need for dietary changes to mitigate environmental damage. Therefore, excessive consumption of ruminant meat is harmful to the environment, and switching to plant proteins can reduce the ecological impact and promote sustainability. Especially in Romania, where meat consumption is high, promoting alternatives and reducing the consumption of beef and mutton can save water and land, contributing to the preservation of the environment [26].

Studies show plant-based diets offer essential nutrients with less environmental cost than meat. Lowering meat consumption helps fight climate change and conserve resources like water and land [17,46–49,66].

Additionally, transitioning to a plant-based protein diet can be successfully implemented in any country, regardless of its available resources. Plant cultivation for protein production generally requires less water and land than animal farming for meat [67,68]. Thus, even countries with limited resources can benefit from this shift in the dietary paradigm.

Furthermore, it is important to emphasize that transitioning to plant protein sources does not only mean a change in diet but also new economic and social opportunities. Plant protein production can be more accessible and sustainable than meat production, thus offering opportunities for development and economic growth in rural communities and developing countries [69–71].

The long-term impact of even modest reductions in meat consumption of 100 g/week underscores the importance of sustainable management of natural resources. Policymakers and stakeholders should consider strategies to incentivize and support dietary transitions towards more environmentally friendly options, ensuring a balance between nutritional needs and ecological preservation [72,73].

While individual choices matter, addressing the complex challenges of food sustainability requires collective action [63]. By fostering a culture of sustainability, empowering individuals with knowledge, and implementing supportive policies, we can collectively mitigate the environmental impact of food consumption and foster a more resilient future.

Therefore, this research supports a global model for reducing meat consumption by just 100 g/week in favor of plant proteins, a practical approach for any country to enhance health, the environment, and dietary sustainability.

The findings underscore the need for ongoing research and dialogue on sustainable food systems [57]. By continuously exploring the interconnectedness of dietary choices, environmental sustainability, and societal well-being, we can form evidence-based policies and empower individuals to make informed decisions for a healthier planet.

It is imperative to take responsibility for protecting natural resources by adopting a benevolent attitude and responsible food consumption at an individual level. Individual dietary choices can significantly impact environmental conservation and food security. Shifting towards plant-based diets reduces our ecological footprint and supports global ecological balance. This change, beneficial both for personal health and the environment, aligns with the broader aim of a sustainable society [2,21,37,73,74].

Therefore, the presented model underscores the balance between personal freedom, environmental protection, and societal duty, suggesting that informed, collective actions can lead to a sustainable food future.

**Author Contributions:** Conceptualization, T.I.T., M.O. and R.G.; methodology, R.A.L. and I.M.B.; software, G.A.F.N.; validation, T.I.T., M.O. and R.G.; formal analysis, I.B. and C.T.; investigation, R.A.L. and I.M.B.; resources, M.O. and R.G.; data curation, M.O. and G.A.F.N.; writing—original draft preparation, T.I.T., M.O. and R.G.; writing—review and editing, I.B. and C.T.; visualization, R.A.L. and I.M.B.; supervision, T.I.T. and I.M.B.; project administration, T.I.T.; funding acquisition, M.O. and R.G. All authors have read and agreed to the published version of the manuscript.

**Funding:** This research was performed in the frame of the project Development and consolidation of the METROFOOD-RI National Network, a grant offered by the Romanian Minister of Research, Innovation, and Digitalization as the Intermediate Body for the Competitiveness Operational Program 2014–2020, call POC/78/1/2/, project number SMIS2014 + 136213, acronym METROFOOD-RO, at the Research Institute for Biosecurity and Bioengineering at ULST, Romania.

**Data Availability Statement:** Data were obtained from the National Institute of Statistics Romania and are available at https://www.recensamantromania.ro/rezultate-rpl-2021/rezultate-definitive/ and https://insse.ro/cms/sites/default/files/field/publicatii/anuarul_statistic_al_romaniei_carte_ed_2023_0.pdf, with open access to the National Institute of Statistics Romania (https://insse.ro/cms/en/content/information-services-0, all accessed on 11 February 2024).

**Conflicts of Interest:** The authors declare no conflicts of interest.

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
