# Peer review of "Synergy between the Waste of Natural Resources and Food Waste Related to Meat Consumption in Romania"

_agriculture, doi:10.3390/agriculture14040644_

Round 1
Reviewer 1 Report
Comments and Suggestions for Authors
Aiming at Romania, this paper focuses on Natural Resources and Food Waste Related to Meat Consumption in Romania, which is very meaningful. Some suggestions are put forward for the author's reference
First, the importance of the research question, and the uniqueness of Romania, need to be supplemented;
Second, in the discussion process, it is necessary to have a more thorough dialogue with the existing literature to show the consistency between this paper and the existing research, the differences, and the shortcomings of the research
Third, the conclusions are too complex and need to be more concise and focused.
Fourthly, the title of the paper is too long, and the subtitle is suggested to be deleted
On the whole, the article is still good, it is recommended to revise and finally publish.
Comments on the Quality of English LanguageThe author should pay attention to the English expression, which needs to be slightly modified
Author Response
Dear Reviewer 1,
Thank you for your time and for your valuable observations and comments, which helped us to improve our manuscript!
We accepted all that you suggested, and we completed and corrected the manuscript accordingly.
Below are our answers, point by point.
Comments 1. First, the importance of the research question, and the uniqueness of Romania, need to be supplemented.
Response 1. As you suggested, we completed the importance of the research question and the uniqueness of Romania, and we also included data, a figure about it, and reference sources.
Comments 2. Second, in the discussion process, it is necessary to have a more thorough dialogue with the existing literature to show the consistency between this paper and the existing research, the differences, and the shortcomings of the research
Response 2. The Discussion section has been revised and completed, as you suggested.
Comments 3. Third, the conclusions are too complex and need to be more concise and focused.
Response 3. We modified the text of the Conclusions, as you suggested.
Comments 4. Fourthly, the title of the paper is too long, and the subtitle is suggested to be deleted
Response 4. We deleted the subtitle, as you suggested.
Comments 5. The author should pay attention to the English expression, which needs to be slightly modified.
Response 5. As you suggested, we have corrected the English expressions and others that were necessary, due to the changes made.

Reviewer 2 Report
Comments and Suggestions for Authors
Synergy between Waste of Natural Resources and Food Waste Related to Meat Consumption in Romania - Seeking Sustainable Path to Responsible Consumption
Agriculture
(Manuscript ID: agriculture-2902276)
This is an interesting and well-written paper that analyzes the contrast between an individual’s ability to choose their diet and the broader consequences of excessive food consumption and waste. This paper specifically investigates the potential advantages of moderate dietary changes - decreases in the consumption of animal products (specifically meat) - on natural resources and the environment. The analysis is implemented using data from Romanian and international sources. The main results show that small reductions in meat consumption could significantly reduce water usage, carbon emissions, and land utilization. These results are used to advocate positive transformation in dietary habits to support environmental preservation and promote health benefits. I have a few comments and suggestions that I list below.
For the unfamiliar reader, the paper could provide some supplementary notes about the water footprint formula (WFRt), the carbon footprint formula (CO2eFRt), and the land use footprint formula (LuFRt).
The paper would benefit from briefly discussing the potentially varying implications of consuming domestically produced and imported meat. Supplementary notes regarding the corresponding agricultural and trade policies could be provided in this context.
I wonder whether the claim “This is a universal solution to global problems, in this case, reducing consumption by just 100g/week of meat, preferably ruminant meat” is heroic. While reducing meat consumption would lead to the preservation of natural resources and the environment, the magnitude of these effects could vary across countries and regions. It could be a function of environmental, economic, and socio-economic conditions. I suggest providing some related discussions and potentially mitigating this claim.
Could the authors provide some supplementary notes (and/or estimates) about the impacts of substituting one type of meat (beef) for another type of meat (poultry) on the water footprint, the carbon footprint, and the land use footprint? In this context, the author(s) could carry out the analysis based on the extent of substitution across meat commodities.
Also, the reader may wonder about the cross-price elasticities across meat commodities and the resulting implications for the consumption of different meat commodities.
It is noted that “examining the real cost of meat in Romania also requires a careful consideration of the social and economic aspects.” I suggest elaborating further on this important point.
Comments on the Quality of English LanguageMinor editing of English language required.
Author Response
Dear Reviewer 2,
Thank you for your appreciation of our manuscript and especially for your valuable observations and comments, which helped us to improve our manuscript!
We accepted all that you suggested, and we completed and corrected the manuscript accordingly.
Below are our answers, point by point.
Comments 1. For the unfamiliar reader, the paper could provide some supplementary notes about the water footprint formula (WFRt), the carbon footprint formula (CO2eFRt), and the land use footprint formula (LuFRt).
Response 1. We have included additional notes on all three formulas.
Comments 2. The paper would benefit from briefly discussing the potentially varying implications of consuming domestically produced and imported meat. Supplementary notes regarding the corresponding agricultural and trade policies could be provided in this context.
Response 2. As you suggested, we have completed the possible variable implications of the consumption of meat produced/imported in Romania and we have provided, in short, additional notes on the Romanian agricultural and commercial policies.
Comments 3. I wonder whether the claim “This is a universal solution to global problems, in this case, reducing consumption by just 100g/week of meat, preferably ruminant meat” is heroic. While reducing meat consumption would lead to the preservation of natural resources and the environment, the magnitude of these effects could vary across countries and regions. It could be a function of environmental, economic, and socio-economic conditions. I suggest providing some related discussions and potentially mitigating this claim.
Response 3. We have included a discussion about the aspects related to the relationship between environmental, economic, and socio-economic conditions and their effects, as you suggested. At the same time, we mitigated the statement that "This is a universal solution to global problems", and we mentioned that this could be one of the potential solutions to global issues concerning environmental footprints, particularly addressing the situation of overconsumption of meat, a situation that Romania finds itself in, by reducing the weekly consumption by just 100g, preferably ruminant meat.
Comments 4. Could the authors provide some supplementary notes (and/or estimates) about the impacts of substituting one type of meat (beef) for another type of meat (poultry) on the water footprint, the carbon footprint, and the land use footprint? In this context, the author(s) could carry out the analysis based on the extent of substitution across meat commodities.
Response 4. Realizing, following this recommendation, that we did not emphasize enough the fact that Romanians' diet is deficient, we introduced data that more obviously emphasizes the fact that Romanians have been consuming, for many years, much more meat than is recommended by authorities in the field of dietetics. Thus, we introduced the text and related figures into the text.
We believe that, according to the data presented, Romanians consume too much meat, more and more from year to year, and replacing it with another type of meat would not bring an important benefit. The important benefit would be the reduction of weekly meat consumption. However, according to your recommendation and considering that there are also people in Romania who do not overconsume meat, we have now included in the manuscript, possible substitutions of beef, both with animal-based and plant-based protein sources.
Regarding the analysis based on the extent of substitution across meat commodities, the impact of other types of meat on the water footprint, carbon footprint, and land use, they are presented in Table 3. Water footprint related to 100 g of different types of meat, Table 5. Carbon footprint related to 100 g of different types of meat, and Table 7. Land use footprint related to 100 g of different types of meat.
Comments 5. Also, the reader may wonder about the cross-price elasticities across meat commodities and the resulting implications for the consumption of different meat commodities.
Response 5. The purpose of this article is to emphasize that a reduction in meat consumption, even a small one, such as the one we proposed of 100 g/week, has a major impact on the environmental footprint, if it is done consistently and by as many consumers as possible. We avoided discussing the issue of prices, precisely in order not to give a pecuniary connotation to the results of the research, but an ethical one. Of course, beef being the most expensive meat in Romania, and much more expensive than other animal or plant-based protein sources, it would have been easy to do this, but we considered that the beneficial impact on the environment could be overshadowed by the beneficial impact on consumer budgets. However, following your recommendation, we included a discussion in manuscript in which we emphasized that by reducing meat consumption or substituting it with other types of animal or plant-based protein sources, consumer budgets will be positively affected because meat in general, but especially beef, is one of the most expensive foods in the Romanian market. We also included a discussion that underlines that given the cross-price elasticities, in the case of substituting beef with another source of protein, there may be a surge in demand for alternative sources of animal-based proteins, particularly chicken meat and pork, leading to their price escalation.
Comments 6. It is noted that “examining the real cost of meat in Romania also requires a careful consideration of the social and economic aspects.” I suggest elaborating further on this important point.
Response 6. As you suggested, we supplemented the manuscript with these aspects, which we presented and argued.
Comments 7. Minor editing of English language required.
Response 7. We have operated the English editing of the text, where it was the case, as you suggested.

Reviewer 3 Report
Comments and Suggestions for Authors
Reducing the environmental burden of food consumption is essential for improving human and ecological sustainability. This study has focused on the environmental benefits of reducing meat consumption in Romania. However, the significant information in this study is adopted from other essential studies, and the authors’ academic contribution is quite limited.
The authors should at least discuss the range of possible differences between the actual size of the environmental burden due to meat consumption in Romania and the estimated size in the study using averaged unit values of water, carbon, and land footprints. Meat production places and transportation distances/modes can cause significant differences between these two values, and thus, this discussion is indispensable.
The health and dietary impact of the weekly reduction in 100 g of meat consumption among Romanian adults should be discussed. The authors may first describe the current health and dietary situations and then discuss the impact of avoiding meat eating.
Alternative diet sources, such as plant-based protein, may be needed in place of reduced meat consumption. The environmental burden of these alternative food sources must be considered when estimating the net impacts of meat avoidance.
The data treatment process should be described in detail. For example, why does the water footprint value in Table 3 not become one-tenth of the corresponding value in Table 2? Did the authors select specific portions of meat?
Author Response
Dear Reviewer 3,
Thank you for your time and for your valuable observations and comments, which helped us to improve our manuscript!
We accepted all that you suggested, and we completed and corrected the manuscript accordingly.
Below are our answers, point by point.
Comments 1. The authors should at least discuss the range of possible differences between the actual size of the environmental burden due to meat consumption in Romania and the estimated size in the study using averaged unit values of water, carbon, and land footprints. Meat production places and transportation distances/modes can cause significant differences between these two values, and thus, this discussion is indispensable.
Response 1. We have completed the manuscript with this observation, respectively, we presented in an extended text that when assessing meat's environmental impact in Romania, it's crucial to recognize that average footprints for water, carbon, and land use may not fully reflect the actual impact due to local production variances and supply chain complexities. Also, we underlined the fact that an amount of Romania's beef is imported, with transportation contributing considerably to the overall carbon footprint. We also showed that the production practices and environmental policies differ across regions, influencing the true size of meat production's environmental footprint, and consequently, average unit values might not accurately represent the actual impact. Also, we included in the manuscript that substituting meat, particularly beef, with alternative protein sources, can benefit both the environment and consumer budgets, as beef is among the most expensive foods in Romania.
Comments 2. The health and dietary impact of the weekly reduction in 100 g of meat consumption among Romanian adults should be discussed. The authors may first describe the current health and dietary situations and then discuss the impact of avoiding meat eating.
Response 2. We completed the manuscript with information and discussion about the obesity and overweight situation in Romania, and the health implications of reducing overconsumption of meat. We emphasized the fact that Romania is a state in which a significant percentage of the population suffers from obesity and overweight, and, at the same time, according to the national and international statistics presented and cited, it is also known that Romanians are big meat consumers, much over the dietary recommendations of national and international authorities.
Comments 3. Alternative diet sources, such as plant-based protein, may be needed in place of reduced meat consumption. The environmental burden of these alternative food sources must be considered when estimating the net impacts of meat avoidance.
Response 3. Realizing, following this recommendation, that we did not emphasize enough the fact that Romanians' diet is deficient, we introduced data that more obviously emphasizes the fact that Romanians have been consuming, for many years, much more meat than is recommended by authorities in the field of dietetics. Thus, we introduced the text and related figures into the text.
We believe that, according to the data presented, Romanians consume too much meat, more and more from year to year, and replacing it with another type of meat would not bring an important benefit. The important benefit would be the reduction of weekly meat consumption. However, according to your recommendation and considering that there are also people in Romania who do not overconsume meat, we have now included in the manuscript, possible substitutions of beef, both with animal-based and plant-based protein sources.
Comments 4. The data treatment process should be described in detail. For example, why does the water footprint value in Table 3 not become one-tenth of the corresponding value in Table 2? Did the authors select specific portions of meat?
Response 4. We have detailed the data treatment process with explanations of all three formulas, for a better understanding of the calculation method.
The water footprint value in Table 3 is becoming one-tenth of the corresponding value in Table 2. In Table 2 – “Total water footprint liters/kg meat” is mentioned as 15415, 4325, 5988, and 10412 for Beef, Chicken Meat, Pork, and Sheep Meat, and in Table 3 “Water footprint l/100g meat/week/capita” is mentioned as 1541.5, 432.5, 598.8, 1041.2 for Beef, Chicken Meat, Pork, and Sheep Meat. We have presented the data identically for the carbon footprint and land use footprint.
The specific portion of meat we mentioned is 100 g.

Round 2
Reviewer 3 Report
Comments and Suggestions for Authors
I would value the authors' efforts to address my previous comments. However, the possible discrepancy between the actual environmental burden and the one calculated using globally averaged unit values is still a big problem. At least some quantitative information on the sizes of this discrepancy should be provided. The lack of this information deteriorates the merit of doing this research in Romania.
For example, the authors may choose one or two commonly consumed brands from domestically produced and imported beef and assess their carbon footprint differences from the globally averaged values considering actual transportation length. This is not a difficult task.
Author Response
Dear Reviewer 3,
Thank you again for the time that you spent with our manuscript!
We accepted what you suggested, and we completed and corrected the manuscript accordingly.
Below is our response.
Comment. I would value the authors' efforts to address my previous comments. However, the possible discrepancy between the actual environmental burden and the one calculated using globally averaged unit values is still a big problem. At least some quantitative information on the sizes of this discrepancy should be provided. The lack of this information deteriorates the merit of doing this research in Romania. For example, the authors may choose one or two commonly consumed brands from domestically produced and imported beef and assess their carbon footprint differences from the globally averaged values considering actual transportation length. This is not a difficult task.
Response. As you suggested, we have completed the manuscript with this observation, and we wrote that a considerable proportion of the beef consumed by Romanians comes from imports, especially from Spain and Italy. This highlights the reality that many of the beef products consumed in Romania are brought from far away, which inevitably leads to an increase in the carbon footprint associated with national consumption.
We showed that to get from Spain, with a starting point such as the capital Madrid, to the capital Bucharest, beef travels an impressive distance of 3,181 km, while from Italy, leaving from the capital Rome to the same destination, the capital Bucharest, freight is shorter, but still significant, at about 1,876 km. Each stage of this freight contributes to the global CO2e footprint of the product, from the farm of origin to the consumer's table.
Also, we showed that long-distance transport, whether by road, rail, air, or sea, adds significant CO2e to each kilogram of imported beef. For example, air transport generates 2,210 g of CO2e for each kilogram of meat transported over a distance of 1,000 km, equivalent to 11.20 km traveled by car, according to the data presented in the newly added Table 9. By comparison, road transport, although less intensive and more energy-efficient than air, contributes 0.105 g CO2e over the same distance and weight, equivalent to 0.55 km of driving. We also added Table 10, and we presented concrete examples, as you suggested. We highlighted the fact that beef is imported in the form of beef carcasses, and not in cut pieces as commonly consumed brands. This practice reflects a global trend, adapting the meat-cutting method to the needs of the local market. Therefore, we calculated the increase in the CO2e footprint, per kg of beef carcass, from the most important three beef 3 sources: Spain, Italy, and Romanian domestic production.
